# Application of LC-MS/MS for the Identification of Drugs of Abuse in Driver’s License Regranting Procedures

**DOI:** 10.3390/ph17121728

**Published:** 2024-12-20

**Authors:** Roberta Tittarelli, Lucrezia Stefani, Leonardo Romani, Federico Mineo, Francesca Vernich, Giulio Mannocchi, Maria Rosaria Pellecchia, Carmelo Russo, Luigi Tonino Marsella

**Affiliations:** 1Laboratory of Forensic Toxicology, Section of Legal Medicine, Social Security and Forensic Toxicology, Department of Biomedicine and Prevention, Faculty of Medicine and Surgery, University of Rome “Tor Vergata”, Via Montpellier 1, 00133 Rome, Italy; lucrezia.stefani@students.uniroma2.eu (L.S.); leonardo.romani.09@students.uniroma2.eu (L.R.); f.mineo@med.uniroma2.it (F.M.); francesca.vernich@uniroma2.it (F.V.); giuliomann24@hotmail.com (G.M.); mrosaria.pellecchia@libero.it (M.R.P.); carmelo.russo.01@students.uniroma2.eu (C.R.); marsella@uniroma2.it (L.T.M.); 2PhD School in Medical-Surgical Applied Sciences, University of Rome “Tor Vergata”, Via Montpellier 1, 00133 Rome, Italy

**Keywords:** hair analysis, driving license, drugs of abuse, DUID, UPLC/MS-MS, multiple drug consumption, forensic toxicology, road accidents, driving fitness evaluation

## Abstract

**Background:** Drugged driving is associated with an increased risk of road accidents worldwide. In Italy, driving under the influence (DUI) of alcohol and drugs is a reason for driving disqualification or revocation of the driving license. Drivers charged with driving under the influence of alcohol and drugs must attend a Local Medical Commission (LMC) to undergo mandatory examinations to regain the suspended license. Our study mainly aims to report on the analysis performed on hair samples collected from 7560 drivers who had their licenses suspended for drugged or drunk driving between January 2019 and June 2024. **Methods:** A rapid, sensitive, and selective method for the determination of ethyl glucuronide in hair by UPLC/MS-MS was developed and fully validated. **Results:** The most frequently detected substances were cocaine (ecgonine methyl ester, norcocaine, and benzoylecgonine) and cannabinoids (Δ9-tetrahydrocannabinol, cannabidiol, and cannabinol), followed by opiates (codeine, morphine, and 6-MAM), methadone (EDDP), and amphetamines (amphetamine, methamphetamine, MDA, MDMA, and MDEA). To perform a more in-depth analysis, we also compared hair color with the drug classes that tested positive. The results showed a significant prevalence of dark hair that tested positive for one or more substances, followed by gray/white hair and light hair. **Conclusions:** Our study provides an interesting and alarming insight into drug exposure in the general population with serious public health threats, discussing the main aspects of hair matrix analysis and focusing on its advantages and reliability in the interpretation of results.

## 1. Introduction

Driving under the influence of drugs (DUID) significantly increases crash risk worldwide. The Global Status Report on Road Safety published in 2023 by the World Health Organization (WHO) shows that an estimated 1.19 million fatal road accidents occurred in 2021 [1]. Although these data are decreasing, driving under the influence of alcohol and psychoactive substances is still responsible for about 10% of road traffic fatalities.

Even today, almost 50 million people a year suffer non-fatal injuries, and several are permanently disabled. In 2019, road traffic fatalities continued to be the leading cause of death for children and young people aged 5–29 [1].

According to data reported by WHO on road traffic injuries, between 2010 and 2018, cannabis-related deaths increased from 9% to 21.5%. This behavior also appears to be more correlated with passenger deaths than road accidents not involving this substance [2].

Moreover, in 2022, the United Nations Office on Drugs and Crime (UNODC) listed benzodiazepines as the main substances identified in 67% of drugged driving cases. The UNODC also recorded an increasing number of traffic accidents related to new psychoactive substances (NPS) use; in particular, synthetic opioids were identified in 21% of cases [3].

Over the last years also in Italy, there has been a rise in the number of road accidents caused by people driving under the influence of drugs.

The latest data collected by the Italian National Statistics Institute (ISTAT) in 2022 showed that 165,889 traffic accidents occurred in Italy (+9.2% compared to the previous year), with 223,475 injured (+9.2%) and 3159 deaths (+9.9%). The number of fatalities was almost stable, slightly lower than in 2019 (−0.4%) [4].

According to the Article no. 187 of the Italian Highway Code (Law no. 285/1992), DUID is a reason for driving disqualification or revocation of the driving license. According to the provisions of art. 119 Presidential Decree no. 495/92, drivers charged with driving under the influence of alcohol and/or drugs must attend a Local Medical Commission (LMC) to undergo mandatory examinations to regain the suspended license [5].

One of the physical requirements prescribed by the LMC to evaluate physical fitness is represented by toxicological analyses, performed to exclude any illicit drug use.

These analyses are generally performed on blood, urine, and/or hair. Blood samples are collected for the evaluation of driving impairment, urine is collected to determine recent drug and/or alcohol use, and hair, providing information over a longer time than urine, is predominantly used for the retrospective investigation of chronic drug abuse and for monitoring drug abstinence [6,7]. Since there is a lack of standardization of these collection procedures, the overall fitness to drive is conventionally entrusted to the LMC [8], based on the anamnestic and medical data, psychiatric examination, and forensic toxicological findings.

In the last decades, non-conventional or alternative matrices such as hair, oral fluid, and sweat have increased their use in forensic toxicology, especially due to the several advantages over traditional biological matrices. In general, their collection is non-invasive and relatively easy to perform, and, in some cases, these samples present a wider detection window [9]. In this context, hair testing is considered a complementary matrix alongside urine and blood to monitor drug intake over a long period, from hours (blood) to days (urine) up to months/years (hair).

Hair is a strong and robust matrix, less affected by adulterants than urine, and provides useful information about a drug addiction history or long-term drug exposure [10]. Head hair grows at 1 cm/month on average (0.6–1.42 cm), depending on the hair type and anatomical location [11]. The window of drug detection is considerably longer than other matrices, ranging up to weeks or even years in relation to the different types of specimens (e.g., head, pubic, underarm, or beard hair) and to the length of the hair strands [11,12].

Several mechanisms for drug incorporation into hair have been proposed, but the accurate processes are still unclear.

One of the most endorsed processes of drug incorporation into hair is the diffusion from blood vessels supplying the follicle between the matrix cells and the end of the keratinizing zone of the hair bulb through passive diffusion with concentration gradients [13].

Other possible mechanisms are excretion by sweat and sebum, as well as passive exposure or environmental contamination (e.g., smoke or physical transfer from contaminated hands handling illicit drugs) [14,15].

Three factors affect drug incorporation: hair pigmentation [16] and molecular physicochemical properties such as lipophilicity and basicity [7,14,15,17].

Dark hair incorporates larger amounts of drugs than less pigmented hair (e.g., blonde hair) [15,18,19,20], and in gray hair, drug concentration can be about 10-fold lower than in pigmented hair [7,9]. Lipophilic and basic molecules are incorporated more easily than polar ones, as the pH gradient that exists between plasma (pH 7.3) and melanocytes/keratinocytes (pH 3–6) is better at promoting the incorporation of alkaline drugs than acidic ones [7,21].

Our study mainly aims to report on the analysis performed by ultra-performance liquid chromatography tandem-mass spectrometry (UPLC-MS/MS) at the Laboratory of Forensic Toxicology of Rome “Tor Vergata” on hair samples collected between January 2019 and June 2024, from people who had their licenses suspended for driving under the influence (DUI) of drugs or alcohol.

We also investigated the prevalence of and trends in drug abuse over the years, with particular attention to age, gender differences, and multiple drug consumption.

## 2. Results

Our study was performed on hair samples collected from 7560 drivers convicted of DUI of alcohol or drug, who attended our laboratory from January 2019 to June 2024 (Figure 1).

Starting in 2019, when *n* = 2103 people attended our laboratory for renewal of their driving license, and after a remarkable decline in new accesses recorded in 2020 (*n* = 926), likely related to the Covid-19 pandemic, an increase was observed in 2021 (*n* = 1347) and in 2022 (*n* = 1437), followed by a slight decrease in 2023 (*n* = 1163).

### 2.1. Epidemiological Data

Most of the drivers who had access to our laboratory were males (87.58%; *n *= 6621); females accounted for 12.42% (*n* = 939) of the observed population.

The mean age of the population (*n* = 7560) was 39.7 years. The mean age for men (*n* = 6621) was 40.2 (minimum mean age 19.5 and maximum mean age 68.1), whereas the mean age for women (*n* = 939) was 36.9 (minimum mean age 21.2 and maximum mean age 50.2).

Out of the 7560 tests performed, *n =* 525 (6.94%) tested positive for one or more illicit substances. Out of the *n =* 525 samples tested positive, in *n* = 13 (2.48%) cases, the hair samples were collected from the underarm, in *n* = 27 (5.14%) cases, from the pubic region, and in *n* = 57 (10.86%), from the chest.

A total of 94.48% (*n* = 496) of the observed population tested positive for one class of substances, while 5.52% (*n* = 29) were polysubstance users. Of the drivers who tested positive, 91.81% were males (*n* = 482), and 8.19% (*n* = 43) were females.

The samples were considered positive or negative on the basis of the cut-off for confirmation analysis recommended by the Society of Hair Testing [12,22] and are reported in Table 1.

### 2.2. Toxicological Results

Data were processed by dividing the observed population into two groups: the first group (A) included data from drivers who tested positive for one drug only (*n* = 496) out of all drivers who tested positive (*n* = 525), while the second group (B) included data from drivers found positive for more than one impairing substance (*n* = 29 out of *n* = 525).

#### 2.2.1. Group A

Group A consisted of *n* = 496 drivers who tested positive for a single illicit substance, and cocaine was the most prevalent detected drug. A total of *n* = 389 (78.43%) tests out of the *n* = 496 cases tested positive for a single substance were positive for cocaine and its metabolites.

Δ9-tetrahydrocannabinol (Δ9-THC) was the second most detected illicit substance (*n* = 68; 13.71%). The number of drivers who tested positive for opiates was *n* = 30 (6.05%), and *n* = 28 were positive for codeine and *n* = 2 for morphine. Methadone was detected in *n* = 6 cases (1.21%). Two subjects (*n* = 2; 0.40%) were positive for the amphetamines class (both amphetamine and methamphetamine), and only one subject tested positive for MDMA (*n* = 1; 0.20%) (Figure 2).

Among females positive for a single drug (*n* = 40), the most frequently detected substance was cocaine (*n* = 28; 70.00%), followed by Δ9-THC (*n* = 11; 27.50%) and opiates (*n* = 1 positive for morphine; 2.50%).

Cocaine was the most prevalent drug also for males, as it was detected in 79.17% of the observed cases (*n* = 361). Δ9-THC was detected in *n* = 57 drivers (12.50%), followed by opiates (*n* = 29; *n* = 27 positive for codeine and *n* = 2 positive for morphine; 6.36%), methadone (*n* = 6; 1.32%), amphetamine (*n* = 2; 0.44%), and MDMA (*n* = 1; 0.22%).

#### 2.2.2. Group B

Group B consisted of *n* = 29 drivers who tested positive for more than one illicit substance (5.52%) out of the drivers who tested positive (*n* = 525).

The most observed association in Group B was between cocaine and Δ9-THC (*n* = 24; 82.76%), followed by cocaine and codeine (*n* = 2; 6.90%). We also observed several other minor associations between MDMA and Δ9-THC (*n* = 1; 3.45%), MDMA and cocaine (*n* = 1; 3.45%), and MDMA, cocaine, and Δ9-THC (*n* = 1; 3.45%) (Table 2).

We also highlighted several differences regarding the sex of the users. The association between two or more substances was significantly higher for males (*n* = 26) than females (*n* = 3).

Males (*n* = 26) were found positive for cocaine and Δ9-THC in 88.46% (*n* = 23) out of the observed cases, followed by cocaine and codeine (*n* = 2; 7.69%) and MDMA and cocaine (*n* = 1; 3.85%).

Females (*n* = 3) were positive for cocaine and Δ9-THC (*n* = 1), MDMA and Δ9-THC (*n* = 1), and the combination of MDMA, cocaine, and Δ9-THC (*n* = 1).

Data obtained from the observation of the two groups (A and B) are summarized in Figure 3.

### 2.3. Hair Color Characteristics

For the interpretation of the distribution of hair color on drivers who tested positive for one or more illicit substances, the samples collected from areas other than the scalp were excluded.

Among hair samples that tested positive (*n* = 428), a strong predominance of brown/black color (dark hair) over gray/white and blonde/red (light hair) was observed, with *n* = 326 (76.17%) positive dark hair, *n* = 74 (17.29%) gray/white hair, and *n* = 28 (6.54%) blonde hair.

To perform a more in-depth analysis, we compared hair color with the drug classes that tested positive. The results showed a significant prevalence of dark hair that tested positive for cocaine (*n* = 230; 53.74%), followed by gray/white hair (*n* = 59; 13.78%) and light hair (*n* = 18; 4.20%).

Δ9-THC was found in *n* = 56 samples of dark hair (13.08%) and *n* = 4 (0.93%) and *n* = 7 (1.63%) samples of gray and light hair, respectively. Opiates (*n* = 24 positive for codeine; *n* = 2 for morphine) have been found in *n* = 17 (3.97%) samples of dark hair, in *n* = 7 (1.63%) gray hair samples, and in *n* = 2 (0.47%) light hair, for a total of *n* = 26 positive hair samples. Amphetamine and MDMA tested positive in *n* = 2 samples of dark hair (0.47%), while methadone tested positive in *n* = 1 sample of dark hair (0.23%) and in *n* = 2 samples (0.47%) of gray hair (Table 3).

We also observed that the most common association of drugs in dark hair was cocaine and Δ9-THC with *n* = 15 cases (3.50%), followed by cocaine and opiates (*n* = 1; 0.23%), MDMA and Δ9-THC (*n* = 1; 0.23%), and MDMA, cocaine, and Δ9-THC (0.23%).

Two drivers (*n* = 2; 0.47%) with gray/white hair and *n* = 1 subject (0.23%) with blonde hair were tested positive for the combination of cocaine and Δ9-THC.

## 3. Discussion

Our main aim was to provide information on illicit substance use, among people who have incurred penalties or traffic violations in relation to the Highway Code and been imposed by the Authorities for DUI of alcohol and/or drugs. We narrowed down the most abused illicit substances in Italy based on the classes of drugs required by Local Medical Commissions (cocaine, cannabinoids, opiates, methadone, amphetamines, and MDMA and their metabolites).

We focused on the demographic patterns of the study population (age, sex, and hair sample characteristics) and the most prevalent drug combination to assess the consumption trend within the selected population and the risk factors associated with impaired driving.

The findings that emerged from our study described the evolution of the phenomenon concerning the abuse of psychotropic substances in the territories belonging to the metropolitan city of Rome.

The analyses were performed on hair, as this matrix provides information on long-term use of substances of abuse allowing the identification of these drugs in a wide diagnostic window. Therefore, we routinely checked hair samples related to the last 3–4 months (4 cm length).

We also focused on hair color because the amount of pigments, such as melanin, can affect drug absorption and incorporation [23].

The deposition of drugs in hair may be also affected by the lipophilicity, polarity, and basicity of the main drug or its metabolites. In general, darkly pigmented hair binds larger amounts of drug than less pigmented hair, because the analytes are assumed to have a more efficient affinity for the melanin pigment present in colored hair. Melanin has both a hydrophobic and acidic nature, which is responsible for the hair pigment’s affinity to alkaline drugs such as cocaine, codeine, and ketamine [18]. Moreover, the products used for bleaching treatments contain strong bases, which may affect the stability or amount of the drug incorporated in hair [15].

Our results showed the highest prevalence of cocaine-positive subjects among the dark-haired group, with *n* = 230 (43.73%) drivers out of the *n* = 525 total hair samples tested positive for illicit substances. Overlapping results were obtained for cannabinoids (dark hair = 56; gray hair = 4; light hair = 7) and opiates (dark hair = 17; gray hair = 7; light hair = 2).

The average age values of drivers who tested positive were 40.2 years for males and 36.9 years for women.

Moreover, according to the National Institute of Statistics (ISTAT), the gender distribution of fatalities showed a distinctly male bias in 2022, particularly for drivers, for whom the proportion of men reached 90%.

The results of our investigations are also consistent with data on road accidents from the Italian National Institute of Statistics, according to which males are significantly more often users of psychotropic drugs than females [4].

Our study showed slightly different results than those from European data. A total of *n* = 389 subjects, compared to the entire study population (*n* = 7560), used cocaine (78.43%), *n* = 68 cannabinoids (13.71%), *n* = 30 opiates (6.05%), *n* = 6 methadone (1.21%), *n* = 2 amphetamines (0.40%), and *n* = 1 MDMA (0.20%).

Conversely, according to the European Union Drugs Agency (EUDA), cannabis is still the most popular and commonly consumed illicit drug of abuse in the European Union (EU). Data show that 22.8 million European adults aged 15–64 consumed cannabis in the last year [24].

Also, for the “Prevalence of Drug Use” report published by the UNODC, there was a prevalence of cannabis use in Europe in 2020 in the population aged 15–64, followed by cocaine, amphetamines (amphetamine and methamphetamine), and ecstasy [25].

A similar scenario has been observed in Spain by Gomez-Talagon et al., who studied the prevalence of alcohol, medicines, and illicit substances among Spanish drivers. The authors observed a clear prevalence of cannabis use, followed by alcohol and cocaine, frequently consumed in combination [26].

After cannabis, and according to the European Drug Enforcement Services, cocaine is the second most widely consumed illicit substance in Europe, as 4.0 million adults aged 15–64 used it [24].

Epidemiological studies on drug abuse behaviors have shown that cocaine could be a gateway to other drugs. In other words, subjects who have used a given substance are more likely to use another drug, especially if that substance is cocaine.

In our study, the most frequent association was between cannabinoids and cocaine (*n* = 24), confirming European evidence, followed by the combination of cocaine and opiates (*n* = 2), MDMA and cannabinoids (*n* = 1), MDMA and cocaine (*n* = 1), and MDMA, cocaine, and cannabinoids (*n* = 1).

Our results also agreed with data reported in 2021 by Tassoni et al., on the concurrent assumption of polydrug use among drivers. In their study, the authors highlighted that a significant part of the examined population (12.15%) engaged in polydrug use with a strong predominance of males over females. The most represented pattern was the abuse of two substances: cocaine and Δ9-THC, followed by cocaine and morphine and morphine and Δ9-THC [6].

Data from the United States also show a trend of the use of cocaine in combination with other substances, such as cannabis, resulting in the enhancement of its harmful effects and increased potential risk of motor vehicle accidents [27,28].

## 4. Materials and Methods

### 4.1. Hair Samples Collection

Hair samples were collected under a strict chain of custody procedure at the Laboratory of Forensic Toxicology, the University of Rome “Tor Vergata”, from drivers convicted of driving under the influence of alcohol or drugs. The tests were mandatory as required by the LMC for fitness to drive, and the data obtained from hair drug testing were aggregated and anonymized, as they were collected for non-medical purposes.

The inclusion criteria focused on drivers, aged older than 18 years, who had a suspended license due to the violation of Articles 186 (Driving under the influence of alcohol), and 187 (Driving under the influence of drugs) of the Italian Highway Code. The exclusion criteria were as follows: subjects under the age of 18, applications for a firearm license, and requests for toxicological analysis from courts or hospitals.

The hair collection, the sample storage and the analyses were carried out according to the guidelines of the Society of Hair Testing (SoHT) and the Italian Group of Forensic Toxicologists (GTFI) [12,29].

Hair samples were collected from the vertex posterior on the back of the head as close as possible to the scalp. The locks were marked at the proximal end, were secured in a bundle with a string to distinguish the alignment, and were stored in an envelope at room temperature and in the dark until analysis [30].

If the hair was too short, two clumps (approximately 200 mg) were collected and divided into two different envelopes.

Maintaining the alignment of the strands, as thick as a pencil, the head hair was divided into two aliquots (aliquots A and B). An adequate amount of hair (not less than 100 mg for each aliquot) was collected to perform both the qualitative and quantitative analyses (aliquot A) and to retain another aliquot for a possible counter-analysis (aliquot B). The analyses were performed on the first 4 cm of scalp hair to investigate a period of 3–4 months prior to the hair collection.

Alternatively, as stated in the guidelines, hairs were collected from other body districts (underarms, chest, pubis), taking care to take an amount of about 200 mg. The hair thus collected was divided into two aliquots (A and B) of approximately 100 mg each.

#### Hair Sample Treatment

After decontamination, 25 mg of hair, added to 5 μL internal standard (IS) working solutions, were finely cut and incubated with 500 μL M3^®^ reagent at a controlled temperature of 100 °C for 60 min. Then, samples were cooled at room temperature, and 1 μL of the supernatant was injected directly into the chromatographic system.

### 4.2. UPLC-MS/MS Analysis

#### 4.2.1. Calibrators and Quality Control Solutions

Stock solutions of each standard at 10 µg/mL were prepared in methanol. Two standard stock solutions were prepared, the first containing cannabinoids and cocaine metabolites and the second all other analytes at 250 ng/mL and 1000 ng/mL, respectively, and they were stored in glass vials at −20 °C.

Internal standard stock solutions were prepared in methanol, the first with cannabinoids and cocaine metabolites deuterated and the second containing all other deuterated analytes, both at 10 µg/mL, and they were stored in glass vials at −20 °C. Calibrator working solutions and quality controls (QCs) were daily prepared from the standard stock solution in methanol (5 calibrators along the working concentration range for hair and 3 levels of quality controls).

#### 4.2.2. Instrumentation

Analysis was performed with a UPLC Acquity H Class (Waters, Milford, MA, USA) equipped with an Atlantis Premier BEH C18 AX (2.5 μm 2.1 × 100 mm) (Waters, Milford, MA, USA) column, set at a temperature of 50 °C. The chromatographic system was interfaced with a tandem quadrupole mass spectrometer (XEVO TQD, Waters, Milford, MA, USA). The chromatographic run lasted for 11 min with a gradient composed of mobile phases A and B produced by Comedical Srl (Trento, Italy), at the flow rate of 0.4 mL/min. Initial conditions were 90:10 (A/B). Phase A was gradually ramped down from 90 to 0% and phase B gradually ramped up from 10 to 100%. Mass spectrometric analysis was performed in positive ion multiple reaction monitoring (ES+ MRM) mode. Two transitions for each analyte and deuterated standards were selected. Transitions, the relative cone voltage (V), and the collision energy (eV) are reported in Table 4 for all the analytes under investigation.

#### 4.2.3. Method Validation

Our method was fully developed and validated in accordance with updated established international criteria [31,32]. The linearity ranged from the limit of quantification (LOQ) to 500 pg/mg for cocaine metabolites and cannabinoids and from the LOQ to 2000 pg/mg for all other analytes. Accuracy, precision, selectivity, linearity, sensitivity, and carryover were calculated by injecting five different daily replicates of the calibration points. Five replicates of quality control (QC) samples were also performed. Dilution integrity was tested for over-the-curve samples with a concentration 10 and 50 times higher than the highest calibrators, with a dilution in the mobile phase before sample treatment, verifying precision and accuracy to be within 15%. The analytical recovery and matrix effect (ME) were determined using the experimental design proposed by Matuszewski et al.: set 1 was composed of 5 replicates of standard analytes diluted in the mobile phase at low, medium, and high QC concentrations; sets 2 and 3 were composed of 5 hair blank samples fortified with analytes after and before extraction, respectively, at the same concentration as the replicates of set 1, for each analyte and concentration. ME was calculated by dividing the mean peak areas of set 2 by set 1, and process efficiency (PE) was calculated by dividing the mean peak areas of set 3 by set 1 [33].

The method presented here allowed us to detect all the target analytes with a run of 11 min after a simple 1 h pretreatment of the samples.

No additional peaks due to endogenous substances and carryover interfering with analytes and ISs were detected. The method was linear for all analytes under investigation, with a determination coefficient (r 2) always better than 0.99. Limits of detection (LODs) ranged from 3.0 to 8.0 pg/mg in hair, while LOQs ranged between 10.0 and 25.0 pg/mg in hair. The PE of analytes under investigation was always better than 70%, and ion suppression due to the matrix effect was within 10%. Intra-assay and inter-assay precision and accuracy were always higher than 15%.

### 4.3. Data Analysis

Data analyses and graph preparation were performed with Microsoft Excel^®^ 2016 MSO (16.0.4738.1000) (Microsoft Corporation^®^, Redmond, WA, USA).

## 5. Conclusions

Our data clearly show how the prevalence of drug use is an ever-expanding phenomenon, especially in the male population and in drivers with an average age in their early 40s. Another concern is the rising incidence of polydrug users, especially cocaine and cannabis. This rising trend represents a serious social and public health problem for themselves and others, requiring information and prevention campaigns, especially in the adult driving population. This phenomenon is indeed strongly correlated with an increased incidence of traffic accidents, mainly caused by visual and driving control deficits due to the use of these substances [34].

To the best of our knowledge, the present study provides interesting epidemiological data over a wide period, which can offer both deep and alarming insights into drug exposure in the general population, discussing the main aspects of hair matrix analysis and focusing on its advantages and reliability in the interpretation of results.

However, the lack of firm data on the binding mechanism of substances of abuse in hair, and the effects of cosmetic treatments on their incorporation, will require further research for the proper interpretation of the analytical data. Moreover, although the results are supported by the literature, the association between hair color and the likelihood of testing positive for one or more substances is still being investigated.

## Figures and Tables

**Figure 1 pharmaceuticals-17-01728-f001:**
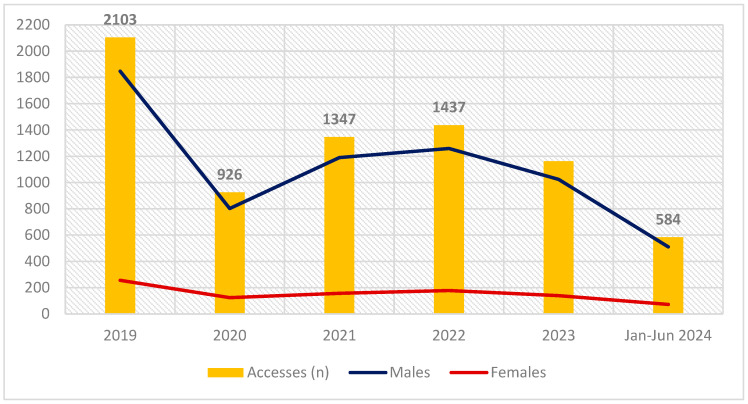
Total number of drivers attending the laboratory for each year under investigation. In blue is shown the clear predominance of males over the number of females, which is shown in red.

**Figure 2 pharmaceuticals-17-01728-f002:**
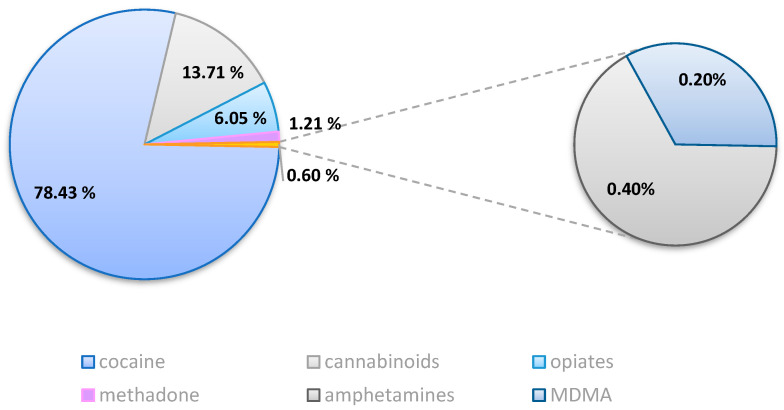
The prevalence of the drugs detected in Group A compared to the total of drivers who tested positive for a single substance (*n* = 496).

**Figure 3 pharmaceuticals-17-01728-f003:**
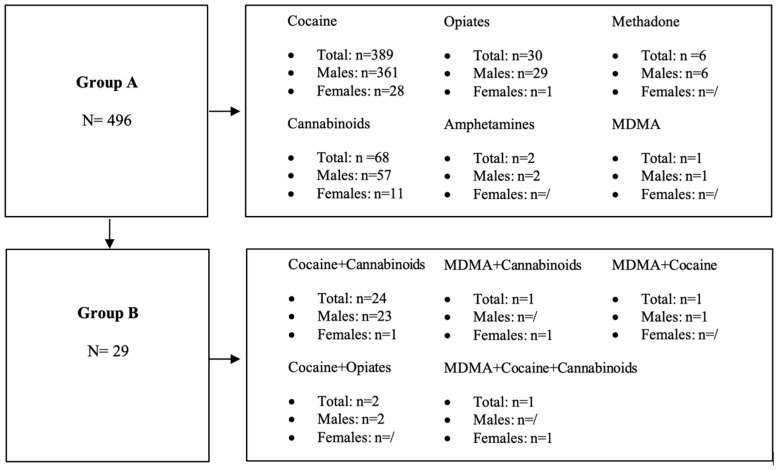
Drivers who tested positive for one (Group A) or more (Group B) illicit substances split by sex.

**Table 1 pharmaceuticals-17-01728-t001:** Analytes and cut-off for confirmation analysis according to the SoHT guidelines [12].

Group	Target Analyte	Cut-Off (pg/mg)	Comments
Cannabinoids	Δ9-THC	500	Detection of THC-COOH strongly supports THC use/intake
	CBD		
Cocaine Group	Cocaine	500	The presence of BZE, NC, CE, hydroxyl-cocaine, or hydroxy-benzoylecgonine must be considered to confirm use. For crack cocaine use, anhydroecgonine methyl ester must be considered
Opiates	Morphine	200	Heroin consumption must be differentiated from codeine or morphine use by the presence of 6-acetylmorphine and/or heroin
	Codeine		
	6-MAM		
Methadone	Methadone	200	Confirmation of EDDP definitively proves the use of methadone
Amphetamine Group	Amphetamine	200	
	Methamphetamine		
	MDA		
	MDMA		
	MDEA		

Legend: BZE: benzoylecgonine; NC: norcocaine; CE: cocaethylene.

**Table 2 pharmaceuticals-17-01728-t002:** Frequency and percentage values of the substances used in a combined pattern by the total of the analyzed subjects (*n* = 525).

Substance	Frequency (*n*)
Cocaine+Δ9-THC	24
Cocaine + Codeine	2
Cocaine + MDMA	1
Δ9-THC + MDMA	1
MDMA + Cocaine + Δ9-THC	1

**Table 3 pharmaceuticals-17-01728-t003:** Hair color distribution in drivers who tested positive for different drug classes.

	Hair Color		
Substances	Dark	Gray	Light
Cocaine	230	59	18
Δ9-THC	56	4	7
Opiates	17	7	2
Amphetamine	2		
Methadone	1	2	
MDMA	2		

**Table 4 pharmaceuticals-17-01728-t004:** Instrumental parameters for analytes under investigation.

Analyte	Retention Time (min)	CV (V)	Quantifier MRM Transitions (*m*/*z*)	CE (eV)	Qualifier MRM Transition (*m*/*z*)	CE (eV)
Ecgonine Methyl Ester-d_3_	0.57	33.00	203.2 > 185.1	17.00	–	–
Ecgonine Methyl Ester	0.57	33.00	200.2 > 82.1	23.00	200.2 > 182.1	17.00
Morphine-d_3_	0.60	35.00	289.2 > 61	28.00	289.2 > 201	40.00
Morphine	0.60	35.00	286 > 165.1	40.00	286 > 153	40.00
Codeine-d_3_	0.88	30.00	303 > 215.1	25.00	303 > 61.1	27.00
Codeine	0.88	30.00	300.1 > 215.1	25.00	300.1 > 199.2	27.00
Amphetamine-d_6_	1.14	20.00	150.1 > 91.1	12.00	–	–
Amphetamine	1.14	20.00	136.1 > 119.1	8.00	136.1 > 91.1	15.00
Methamphetamine-d_5_	1.20	20.00	154.8 > 91.8	12.00	154.8 > 121.1	10.00
Methamphetamine	1.20	20.00	150.1 > 91.1	12.00	150.1 > 119.1	10.00
MDA-d_5_	1.23	20.00	185.1 > 110	26.00	185.1 > 163.1	20.00
MDA	1.23	20.00	180.1 > 133.1	18.00	180.1 > 163.1	10.00
6-MAM-d_3_	1.25	30.00	331 > 61.1	30.00	331 > 195.1	36.00
6-MAM	1.25	30.00	328.1 > 165.1	40.00	328.1 > 181.2	40.00
MDMA-d_5_	1.28	20.00	199.1 > 165.1	12.00	199.1 > 135.2	20.00
MDMA	1.28	20.00	194.1 > 163	12.00	199.1 > 135.2	20.00
MDEA-d_5_	1.51	20.00	213.1 > 163.1	14.00	213.1 > 105.1	26.00
MDEA	1.51	20.00	208.1 > 163.2	14.00	208.1 > 135.1	14.00
Cocaine-d_3_	2.06	30.00	307 > 184.7	20.00	307 > 84.8	30.00
Cocaine	2.06	30.00	304.2 > 182.2	20.00	304.2 > 82.3	28.00
Norcocaine-d_3_	2.36	35.00	304.2 > 182.2	20.00	304.2 > 82.3	28.00
Norcocaine	2.36	30.00	290.2 > 136.1	20.00	290.2 > 168.2	14.00
Benzoylecgonine-d_3_	3.10	30.00	293.1 > 171.1	20.00	293.1 > 105.1	32.00
Benzoylecgonine	3.11	30.00	290.1 > 168.1	20.00	290.1 > 105.1	33.00
EDDP-d_3_	3.31	30.00	281.3 > 235.1	30.00	281.3 > 250.2	22.00
EDDP	3.31	45.00	278.2 > 234.2	26.00	278.2 > 186.2	35.00
Methadone-d_3_	4.11	30.00	313.3 > 268.2	14.00	–	–
Methadone	4.11	30.00	310.3 > 265.2	14.00	310.3 > 105.1	32.00
CBD-d_3_	6.61	40.00	318.3 > 123.2	32.00	318.3 > 196.2	22.00
CBD	6.62	35.00	315.15 > 123.1	32.00	315.1 > 193.1	20.00
CBN-d_3_	6.81	40.00	314.3 > 241.2	16.00	314.3 > 223.2	18.00
CBN	7.13	30.00	311.2 > 241	20.00	311.2 > 223	20.00
Delta-9-THC-d_3_	7.35	35.00	318.2 > 123	34.00	318.2 > 196.1	22.00
Delta-9-THC	7.35	35.00	315.21 > 123	34.00	315.21 > 193.1	22.00

Legend: CV: cone voltage; CE: collision energy; 6-MAM: 6-*O*-Monoacetylmorphine MDA: 3,4-Methylenedioxyamphetamine; MDMA: 3,4-Methylenedioxymetamphetamine; MDEA: 3,4-Methylenedioxy-*N*-ethylamphetamine; EDDP: 2-Ethylidene-1,5-dimethyl-3,3-diphenylpyrrolidine.

## Data Availability

The data presented in this study were obtained from the included studies and are openly available.

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
