# Peer review of "Application of LC-MS/MS for the Identification of Drugs of Abuse in Driver’s License Regranting Procedures"

_pharmaceuticals, 2024, doi:10.3390/ph17121728_

Round 1
Reviewer 1 Report
Comments and Suggestions for Authors
The paper addresses the longstanding issue of driving under the influence of drugs (DUID) and uses hair testing as a method for detecting drug use. It should be noted that hair testing is still considered a complementary test alongside urine and blood. Hair analysis offers advantages for assessing drug use over a longer period, which is beneficial for evaluating drivers whose licenses have been suspended.
Introduction Section: The section "1.1. Forensic toxicological hair collection and analysis" should be summarized and included as a paragraph in the introduction. This basic information can be referenced for readers to explore further, rather than being a part of the main paper.
Terminology and Case Numbers: The paper should clarify that the number of cases is low, which limits the ability to draw definitive conclusions about hair color characteristics. This was not a controlled study but rather a screening, so statements about dark or gray/white hair should be made cautiously.
Results Presentation: The results are repeated in the text, tables, and discussion. This redundancy should be minimized to improve clarity.
Additional Data: Including urine and blood results before and after license suspension would provide a more comprehensive view of the cases and correlate with hair analysis findings.
References and Paragraphs: Many paragraphs are short and lack references. Ensure that all statements are well-supported by relevant literature.
Discussion Section: The discussion should be more concise and focused on comparing the findings with other studies.
Author Response
Comment 1: The paper addresses the longstanding issue of driving under the influence of drugs (DUID) and uses hair testing as a method for detecting drug use. It should be noted that hair testing is still considered a complementary test alongside urine and blood. Hair analysis offers advantages for assessing drug use over a longer period, which is beneficial for evaluating drivers whose licenses have been suspended.
Response 1: Thank you for pointing this out. We clarified this issue at the lines 76-78.
Comment 2: Introduction Section: The section "1.1. Forensic toxicological hair collection and analysis" should be summarized and included as a paragraph in the introduction. This basic information can be referenced for readers to explore further, rather than being a part of the main paper.
Response 2: Thank you for your observation. Accordingly to your comment, we summarized the introduction, including the section 1.1 to the main paragraph.
Comment 3: Terminology and Case Numbers: The paper should clarify that the number of cases is low, which limits the ability to draw definitive conclusions about hair color characteristics. This was not a controlled study but rather a screening, so statements about dark or gray/white hair should be made cautiously.
Response 3: Thank you for your comment. We added a clarification about the interpretation of our results at the lines 451-453 in the Conclusions section.
Comment 4: Results Presentation: The results are repeated in the text, tables, and discussion. This redundancy should be minimized to improve clarity.
Response 4: Thank you for your precious observation. We deleted some tables that were pleonastic. You'll find these changes in the document in review mode.
Comment 5: Additional Data: Including urine and blood results before and after license suspension would provide a more comprehensive view of the cases and correlate with hair analysis findings.
Response 5: Dear reviewer thank you for your comment, but we have no information about past toxicological analyses.
Comment 6: References and Paragraphs: Many paragraphs are short and lack references. Ensure that all statements are well-supported by relevant literature.
Response 6: We agree with your comment so we included the paragraph 2.3.1 to the previous one (2.3).
Comment 7: Discussion Section: The discussion should be more concise and focused on comparing the findings with other studies.
Response 7: Thank you for your observation. We summarized the Discussion section as you suggested, removing some data that were already presented in the "Results" section.
Reviewer 2 Report
Comments and Suggestions for Authors
The work of Tittarell et al. is highly relevant and of interest.
The study provides a comprehensive analysis of drugged driving trends in Italy, focusing on hair sample testing from 7,560 individuals who had their licenses suspended for driving under the influence (DUI) between January 2019 and June 2024. It highlights the alarming connection between drug use and increased road accidents, emphasizing the public health implications.
Key findings reveal that cocaine and cannabinoids are the most commonly detected substances, followed by opiates, methadone, and amphetamines. The research also uncovers intriguing correlations between drug positivity and hair color, with dark-haired individuals showing the highest prevalence, followed by gray/white and light-haired individuals. These observations underscore the advantages of hair matrix analysis for its reliability and depth in detecting long-term drug exposure.
The study successfully identifies trends in drug abuse over time, shedding light on differences across age, gender, and patterns of multiple drug use. It offers a sobering perspective on the scale of drugged driving and reinforces the importance of stringent measures like mandatory medical evaluations for license reinstatement. This work is a critical contribution to understanding the intersection of drug use and public safety.
The introduction is relevant and the method description is robust. I have no objections and consider the present version as suitable for publication.
I would also like to congratulate the authors for a work well done.
Author Response
Dear Reviewer, thank you for your comment. We really appreciate your report because behind every manuscript is a lot of effort and work, so the fact that it is appreciated encourages us to do better and better. Thank you.
Reviewer 3 Report
Comments and Suggestions for Authors
The subject of the article is important, and the introduction is very well written. However, the methodology requires complete improvement in one aspect – the authors did not use statistical methods in their study. Therefore, the concept of the work should be reconsidered, appropriate statistical tools should be selected and applied. Then, the results of this statistical analysis should be included in Results and Discussion.
Author Response
Dear reviewer, Thank you for your valuable observations. In our case and for our purposes, we applied only descriptive statistics to determine the existence of hypothetical risk factors (e.g., sex and/or age) in drug users. It was not our goal to make further comparisons or statistical insights of other nature.
Round 2
Reviewer 1 Report
Comments and Suggestions for Authors
Although the authors have made significant changes that have greatly improved the manuscript, the introduction remains lengthy and includes a lot of basic information that is not necessary for this type of article. I would accept this paper with minor corrections, specifically recommending that the introduction be shortened and summarized, particularly from lines 85-127.
Author Response
Comment 1: Although the authors have made significant changes that have greatly improved the manuscript, the introduction remains lengthy and includes a lot of basic information that is not necessary for this type of article. I would accept this paper with minor corrections, specifically recommending that the introduction be shortened and summarized, particularly from lines 85-127.
Response 1: thank you for your precious observation. According to your suggestions, we summarized and shortened the introduction at lines 85-127 as you indicated.
Reviewer 3 Report
Comments and Suggestions for Authors
The authors corrected the manuscript. In my opinion in can now be published in present form.
Author Response
Dear reviewer,
thank you very much for your kind reply and for appreciating our research.